# Incidence and Severity of COVID-19 in Relation to Anti-Receptor-Binding Domain IgG Antibody Level after COVID-19 Vaccination in Kidney Transplant Recipients

**DOI:** 10.3390/v16010114

**Published:** 2024-01-12

**Authors:** A. Lianne Messchendorp, Jan-Stephan F. Sanders, Alferso C. Abrahams, Frederike J. Bemelman, Pim Bouwmans, René M. A. van den Dorpel, Luuk B. Hilbrands, Céline Imhof, Marlies E. J. Reinders, Theo Rispens, Maurice Steenhuis, Marc A. G. J. ten Dam, Priya Vart, Aiko P. J. de Vries, Marc H. Hemmelder, Ron T. Gansevoort

**Affiliations:** 1Department of Nephrology, University Medical Center Groningen, University of Groningen, 9713 GZ Groningen, The Netherlands; 2Department of Nephrology and Hypertension, University Medical Center Utrecht, 3508 GA Utrecht, The Netherlands; 3Division of Nephrology, Department of Internal Medicine, Amsterdam University Medical Center, Location Amsterdam Medical Center, 1105 AZ Amsterdam, The Netherlands; 4Department of Internal Medicine, Division of Nephrology, Maastricht University Medical Center, 6229 HX Maastricht, The Netherlands; 5CARIM School for Cardiovascular Disease, University of Maastricht, 6211 LK Maastricht, The Netherlands; 6Department of Nephrology, Maasstad Hospital, 3079 DZ Rotterdam, The Netherlands; 7Department of Nephrology, Radboud University Medical Center, 6525 GA Nijmegen, The Netherlands; 8Erasmus MC Transplant Institute, Nephrology and Transplantation, Department of Internal Medicine, University Medical Center Rotterdam, 3015 GD Rotterdam, The Netherlands; 9Department of Immunopathology, Sanquin Research, 1006 AD Amsterdam, The Netherlands; 10Landsteiner Laboratory, Amsterdam University Medical Center, University of Amsterdam, 1012 WP Amsterdam, The Netherlands; 11Department of Internal Medicine, Canisius Wilhelmina Hospital, 6532 SZ Nijmegen, The Netherlands; 12Department of Clinical Pharmacy and Pharmacology, University Medical Center Groningen, 9713 GZ Groningen, The Netherlands; 13Leiden University Medical Center, Department of Nephrology and Leiden Transplant Center, 2333 ZA Leiden, The Netherlands

**Keywords:** COVID-19, COVID-19 vaccination, kidney transplantation, efficiency

## Abstract

Kidney transplant recipients (KTRs) elicit an impaired immune response after COVID-19 vaccination; however, the exact clinical impact remains unclear. We therefore analyse the relationship between antibody levels after vaccination and the risk of COVID-19 in a large cohort of KTRs. All KTRs living in the Netherlands were invited to send a blood sample 28 days after their second COVID-19 vaccination for measurement of their IgG antibodies against the receptor-binding domain of the SARS-CoV-2 spike protein (anti-RBD IgG). Information on COVID-19 was collected from the moment the blood sample was obtained until 6 months thereafter. Multivariable Cox and logistic regression analyses were performed to analyse which factors affected the occurrence and severity (i.e., hospitalization and/or death) of COVID-19. In total, 12,159 KTRs were approached, of whom 2885 were included in the analyses. Among those, 1578 (54.7%) became seropositive (i.e., anti-RBD IgG level >50 BAU/mL). Seropositivity was associated with a lower risk for COVID-19, also after adjusting for multiple confounders, including socio-economic status and adherence to COVID-19 restrictions (HR 0.37 (0.19–0.47), *p* = 0.005). When studied on a continuous scale, we observed a log-linear relationship between antibody level and the risk for COVID-19 (HR 0.52 (0.31–0.89), *p* = 0.02). Similar results were found for COVID-19 severity. In conclusion, antibody level after COVID-19 vaccination is associated in a log-linear manner with the occurrence and severity of COVID-19 in KTRs. This implies that if future vaccinations are indicated, the aim should be to reach for as high an antibody level as possible and not only seropositivity to protect this vulnerable patient group from disease.

## 1. Introduction

Kidney transplant recipients (KTRs) are at an increased risk for severe COVID-19. The rate of hospital admissions in this patient group was high and their mortality risk was described to be around 20 percent during the first waves of the pandemic, which was 3- to 4-fold higher compared to matched subjects form the general population [1,2]. Unfortunately, the immunogenicity of COVID-19 vaccination in KTRs is severely impaired, with seropositivity rates of only 3–59% after two mRNA vaccine dosages [3,4]. Several studies described the efficacy of additional COVID-19 vaccinations to increase the percentage of seropositive KTRs [5]. Despite such repeated vaccinations, however, the percentage of KTRs that remains seronegative is considerable and ranges between 24 and 61% after a third vaccination [6,7,8,9]. While this percentage decreases with additional fourth and fifth vaccinations [10], the patients who become seropositive after vaccination often have considerably lower antibody levels compared to the general population [11]. However, the exact clinical impact of this impaired humoral response remains unclear.

Real-life data from large registry studies showed limited vaccine effectiveness in solid organ transplant recipients [12,13] and therefore question the efficacy of vaccination in protecting against COVID-19 in individual transplant recipients.

Importantly, these registry studies do not have information on antibody level after vaccination. Therefore, the question of whether an impaired humoral response after vaccination is associated with infection risk and higher disease burden remains unanswered. Although COVID-19 is currently considered not to be a pandemic anymore, it remains important to address this question for future vaccinations in case of new pandemic urgencies. Knowledge on the relationship between an impaired immune response and infection risk and disease burden will help us to better identify patients still at high risk after vaccination. This may result in additional protective interventions. In this study, we therefore analyse the impact of antibody levels after COVID-19 vaccination on subsequent COVID-19 incidence and severity in a large cohort of COVID-19 naïve KTRs.

## 2. Materials and Methods

### 2.1. Setting and Subjects

KTRs were prioritized for the first two COVID-19 vaccinations in April and May 2021 in the Netherlands. During that period, all adult patients with a functioning kidney transplant were asked to participate in a study for antibody measurement after COVID-19 vaccination. Prior to the start of the national vaccination campaign, a subset of KTRs was invited to participate in a study investigating immunogenicity after COVID-19 vaccination (RECOVAC-IR study [14]) from which data were included in the current study. Ethical approval was obtained from the central ethics committee at the UMC Groningen (METc 2021/099 and METc 2020/662). These studies are registered at www.clinicialtrials.gov (NCT04841785 and NCT04741386).

### 2.2. Data Collection

A detailed description of the study design has previously been published [15]. In short, patients were sent a package to collect a blood sample at home approximately 28 days after their second COVID-19 vaccination (allowed range of 14 to 56 days). With a finger-prick method, five drops of capillary blood were obtained. A questionnaire about demographics, disease history, COVID-19 vaccination, and adherence to COVID-19 measures and restrictions after vaccination was completed. A detailed description of this last questionnaire is added to the Appendix A. Information on socio-economic status was obtained from publicly accessible data from Statistics Netherlands (CBS) [16]. These data contain a score per postal code of households in the Netherlands based on financial prosperity, educational level, and recent employment history of households. The higher this score, the higher the socio-economic status. Additional information on patient characteristics was extracted from the Dutch Organ Transplant Registry (NOTR). At 6 months after vaccination, another questionnaire was completed including questions on the occurrence and severity of COVID-19 during that period. A detailed description of data collection from included subjects of the RECOVAC-IR study (*n* = 298), which encompasses a total follow-up time of 6 months after COVID-19 vaccination, was stated previously [17].

### 2.3. SARS-CoV-2-Specific Antibodies

Sanquin Research and Lab Services (Amsterdam, the Netherlands) analysed the blood samples for the presence of antibodies against the receptor-binding domain (RBD) of the SARS-CoV-2 spike protein (anti-RBD IgG antibody) using an in-house-developed anti-SARS-CoV-2 RBD IgG ELISA assay (cut-off level for seropositivity: ≥50 Binding Antibodies Units (BAU)/mL) [18]. This is an indirect ELISA using microtiter plates coated with RBD and detection by monoclonal mouse anti-human IgG. This assay was combined with an in-house anti-SARS-CoV-2 Nucleocapsid Protein (NP) bridging ELISA, which is an indirect ELISA using microtiter plates coated with NP and detection by biotin-labelled NP [19] (cut-off level for seropositivity: ≥0.17 normalized optical density (nOD)).

### 2.4. Outcomes

Our primary endpoint, COVID-19 after vaccination, was defined by a self-reported positive COVID-19 test (PCR or home-based self-test) after blood sampling for antibody measurement and before a possible third vaccination. Information on whether a patient died was obtained from family members responding to our correspondence or from mortality data from the Dutch Organ Transplant Registry (NOTR). In case of a self-reported positive COVID-19 test, both the patient and their treating physician were contacted to provide additional information on disease severity and treatment. COVID-19 disease severity was categorized using the WHO COVID-19 Clinical Progression Scale (CPS) [20]. Severe COVID-19 was defined as hospitalization and/or death, i.e., a WHO CPS score of 4 or higher (Appendix A). In case of death, the treating physician was contacted to provide details on the cause of death and, in case of COVID-19-related mortality, to provide details on treatment and disease course.

Seroresponse after vaccination was defined as an anti-RBD IgG antibody level ≥ 50 (seropositive) or <50 BAU/mL (seronegative) [18].

### 2.5. Statistical Analysis

For this analysis, we included patients with complete information on the date of their second vaccination, the date of blood sample collection, COVID-19 status at baseline and during follow-up, and successful measurement of antibody level. Patients were excluded if blood was obtained less than 14 days or more than 56 days after the second vaccination or if they were diagnosed with COVID-19 before their blood was obtained. The latter was assessed with a positive NP antibody measurement or using a self-reported previous COVID-19 diagnosis.

Continuous data are presented as a mean with standard deviation (SD) in the case of normal distribution or as a median with inter-quartile range (IQR) in the case of non-normal distribution. Categorical data are presented as absolute numbers and percentages. Comparisons were made using a *t*-test for continuous variables with a normal distribution, the Mann–Whitney U test for non-normally distributed variables, or Pearson’s chi-squared test for categorical variables.

Cumulative incidence curves of COVID-19 were presented by seroresponse after vaccination. Patient follow-up started at the date of blood collection for measurement of anti-RBD IgG. Observations were censored at six months of follow-up, at time of withdrawal of consent, at time of death, or at time of third COVID-19 vaccination, whichever came first.

The hazard ratio (HR) (95% confidence interval) for the association between incidence of COVID-19 and seroresponse after vaccination and incidence of COVID-19 was estimated using Cox-proportional hazards regression analyses. We first analysed the association crude and subsequently after adjustment for age, sex, characteristics that were significantly different between KTRs with and without COVID-19 at follow-up, adherence to COVID-19 restrictions, and social-economic status, which are important confounders for contracting COVID-19 [21,22]. Cases were deleted listwise when covariates were missing. Lastly, to investigate which variables contribute most to the risk of COVID-19, we performed a multivariable stepwise regression analysis. The proportional hazards assumption was investigated by visual inspection of Schoenfeld residuals and investigation of the interaction between follow-up time and individual covariates.

The association between COVID-19 severity (a WHO CPS score ≥ versus < 4) and seroresponse was estimated using a logistic regression analysis. We first tested the association between COVID-19 severity and various variables in univariable analyses that could be associated with COVID-19 severity. The included variables were seroresponse, age, sex, and characteristics that were significantly different between patients with severe COVID-19 (a WHO CPS score ≥ 4) and non-severe COVID-19 (a WHO CPS score < 4) (Appendix A). Cases were deleted listwise when covariates were missing. To investigate which variables contribute most to the development of severe COVID-19, we subsequently performed a multivariable stepwise backward regression analysis.

Several additional analyses were performed to assess the robustness of our findings. First, we repeated our analyses with seroresponse after vaccination expressed according to four categories of increasing anti-RBD IgG antibody levels and second as a continuous variable. Lastly, as an explorative analysis, we compared the COVID-19-related mortality, hospitalization and in-hospital mortality rate in unvaccinated and vaccinated KTRs. For figures in unvaccinated patients, we used data, specifically from the Netherlands, from the second COVID-19 wave (July 2020–April 2021), the period just before the present study period, when vaccination was not yet available, collected in ERACODA (the European Renal Association COVID-19 Database) [23]. We subsequently estimated the vaccine efficacy by calculating the relative risk reduction in COVID-19-related mortality and hospitalization: (1 − (rate in vaccinated patients/rate in unvaccinated patients)) × 100%.

All analyses were performed with IBM SPSS statistics version 28.0 (SPSS Inc., Chicago, IL, USA). Graphs were created with GraphPad Prism version 9.1.0 (GraphPad Software, San Diego, CA, USA) and Rstudio version 1.4.1106 (Rstudio, PBC, Boston, MA, USA). A two-sided *p*-value < 0.05 was adopted to denote statistical significance.

## 3. Results

### 3.1. Baseline Characteristics

From April 2021 to July 2021, all 12,159 KTRs in the Netherlands were invited for antibody measurement after COVID-19 vaccination. Of these, 2885 were included in the present analyses (Figure 1). Baseline characteristics of these participants are presented in Table 1. The median (IQR) anti-RBD IgG antibody level in KTRs was 72.5 (10.7–600) BAU/mL, with a seropositivity rate of 54.7%. Participants who became seropositive after vaccination (*n* = 1578) were significantly younger, had less co-morbidities, had a higher eGFR, had their transplantation a longer time ago, and used a lower number of immunosuppressive agents compared to participants who remained seronegative (*n* = 1307). Overall, participants adhered tightly to restrictions after vaccination, with a median score of 4.25 on a five-point Likert scale, with participants who became seropositive after vaccination adhering less tightly to restrictions than those who remained seronegative. Participants who were excluded for analyses (*n* = 943) were significantly younger, had more often diabetes mellitus; were less often female, Caucasian, and vaccinated with mRNA-1273; and adhered less tightly to restrictions compared to participants that were included for analyses (Appendix A).

### 3.2. Incidence of COVID-19 in Seropositive versus Seronegative Participants

COVID-19 was diagnosed in 62 participants at a median of 172 days after COVID-19 vaccination during a follow-up of 6 months after blood collection. In patients not diagnosed with COVID-19 (*n* = 2823), median follow-up was 151 (138–168) days. Infections took place between 26 April and 6 December 2021, when the delta variant of SARS-CoV-2 was the dominant variant in the Netherlands (Appendix A). Apart from seroconversion rate and antibody levels, there were differences in occurrence of diabetes mellitus and azathioprine use in participants diagnosed with and without COVID-19 (Appendix A).

In seropositive participants, 27 out of 1578 were diagnosed with COVID-19 as compared to 35 out of 1307 seronegative participants (1.7 vs. 2.7%, respectively). The risk for COVID-19 was significantly decreased in seropositive participants as compared to seronegative participants (Figure 2A, Table 2). In a model adjusting for age, sex, diabetes mellitus, azathioprine use, adherence to restrictions, and socio-economic status, a seropositive status at 28 days after COVID-19 vaccination was associated with an even stronger decreased risk for COVID-19 (Model 3, Table 2). After performing the stepwise backward analysis, seroresponse was left as one of the variables that contributes to the risk for COVID-19 (Model 4, Table 2).

When split according to categories of increasing antibody levels, the risk for COVID-19 decreased stepwise from the lowest to the highest category (Figure 2B). Figure 2C depicts the overall SARS-CoV-2 infection incidence in the Netherlands in the general population during the same period as the present study. This demonstrates that the rise in cumulative COVID-19 incidence accelerates when infection incidence in the general population increases.

**Figure 2 viruses-16-00114-f002:**
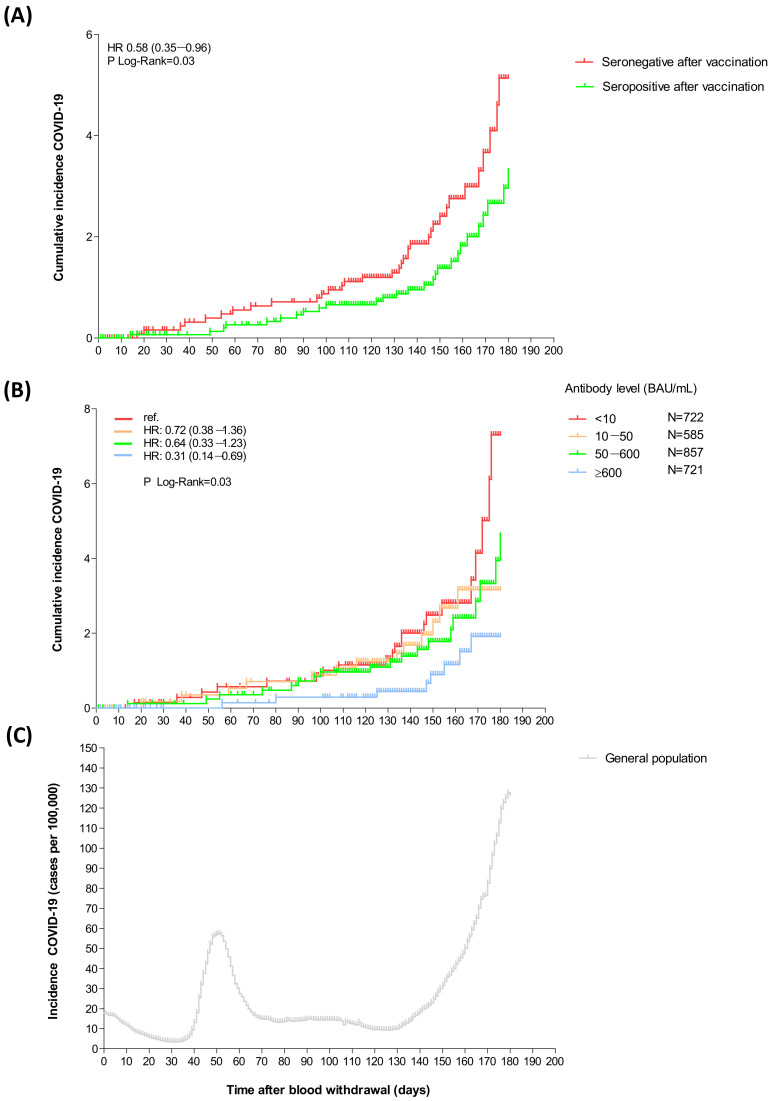
Cumulative incidence of COVID-19 after vaccination for kidney transplant recipients (**A**) who were seronegative or seropositive after vaccination, (**B**) according to categories of increasing antibody level after vaccination and (**C**) incidence of COVID-19 in the general population in the Netherlands (either vaccinated or unvaccinated) during the same period (from 28 May 2021 until 24 November 2021)—numbers were retrieved from freely online accessible data from ‘Our World in Data’ [24]. Hazard ratio (HR) (95% CI) and corresponding *p*-values were calculated using Cox regression analysis.

When studied on a continuous scale, we observed a log-linear relationship between antibody level and risk for COVID-19 without indication for a threshold (HR 0.72 (0.58–0.91) per tenfold higher anti-RBD IgG antibody level, *p* = 0.005) (Figure 3). The association between the risk of contracting COVID-19 and antibody level remained after adjusting for age, sex, diabetes mellitus, azathioprine use, adherence to restrictions, and socio-economic status (HR 0.70 (0.52–0.93), *p* = 0.01). Additionally, antibody level was left as one of the variables that contributes to the risk of contracting COVID-19 after performing a stepwise backward regression analysis.

### 3.3. COVID-19 Disease Severity

Overall, median disease severity was 2 according to the WHO CPS score, and three participants died. In seropositive participants, disease was significantly less severe according to the WHO CPS score than in seronegative participants (2 (2–2) vs. 2 (2–5) *p* = 0.008), as also reflected since none of the seropositive participants were treated with dexamethasone versus 31.4% of seronegative participants (*p* = 0.002) (Table 3). Participants with severe COVID-19 had a lower eGFR, were more often in their first year after transplantation, received a living kidney transplant less often, and had a different socio-economic status compared to participants with non-severe COVID-19 (Appendix A).

Seropositivity was a univariable associated with a lower risk of hospital admission and/or death and was left as one of the contributing variables after performing a multivariable stepwise backward regression analysis (Table 4). When studied on a continuous scale, we observed that participants with higher antibody levels had a lower risk of hospital admission and/or death (OR 0.42 (0.22–0.81) per tenfold higher anti-RBD IgG antibody level, *p* = 0.01), and like seroresponse, antibody level was left as a contributing variable after performing the stepwise backward regression analysis.

### 3.4. Estimated COVID-19 Vaccine Efficacy

In the time period immediately before vaccination, the overall COVID-19 mortality and hospitalization rates in unvaccinated KTRs in the Netherlands were higher compared to the rates in vaccinated KTRs of this study (19.4% vs. 6.5% *p* = 0.03 and 56.3% versus 24.2%, *p* < 0.001, resp.), whereas the in-hospital COVID-19 mortality rate was similar (30.9% vs. 26.7%, respectively, *p* = 0.73) (Appendix A). The stratification of vaccinated patients according to seroresponse revealed that hospitalization and COVID-19 mortality rates were also lower in vaccinated seronegative compared to unvaccinated patients, although the latter did not reach formal statistical significance. Seropositivity after vaccination in KTRs resulted in an 80.9% lower COVID-19-associated mortality risk and a 86.9% lower risk of hospital admission. In seronegative vaccinated patients, these percentages were estimated to only be 55.7 and 34.1%, respectively.

## 4. Discussion

In this study, we demonstrated that seropositivity at 28 days after COVID-19 vaccination is independently associated with a lower incidence and a less severe disease course of COVID-19 in KTRs. Importantly, our data indicated that there is a log-linear relationship between incidence and severity of COVID-19 and vaccination-induced anti-RBD IgG antibody levels, without a threshold above which optimal protection was observed.

The effectiveness of COVID-19 vaccines in reducing the SARS-CoV-2 infection rate and lowering COVID-19 severity and mortality has been uniformly demonstrated in the general population [25]. However, conflicting results were reported specifically in KTRs, which is a vulnerable patient group at high risk of adverse outcomes after SARS-CoV-2 infection. Several studies demonstrated lower immunogenicity after COVID-19 vaccination [4,26,27,28,29,30], but only a few studies investigated whether vaccination is effective in lowering COVID-19 incidence and severity in KTRs [31,32]. Hamm et al. [33] showed the protective effects of COVID-19 vaccination in a national registry study from Denmark that included 1428 solid organ transplant recipients, of whom the majority were KTRs. COVID-19-related hospitalization rates were lower in vaccinated compared to unvaccinated patients (26.4% versus 48.5%, *p* = 0.01), as was COVID-19-related mortality (1.8% versus 9.1%, *p* = 0.047). On the other hand, Callaghan et al. [13] could not corroborate the clinical effectiveness of vaccination in solid organ transplant recipients in a study that included 43,481 subjects by linking four national registries in England. They could not demonstrate a lower infection risk in vaccinated versus non-vaccinated patients. However, their data did indicate a marginally reduced risk of death in vaccinated solid organ transplant recipients [13]. Lastly, a registry study from Ontario in Canada that included 12,842 solid organ transplant recipients showed limited vaccine effectiveness after first and second vaccination, but adjusted vaccine effectiveness against infection improved to 72% after a third dose [12]. Taken together, these data from registry studies question the effectiveness of vaccination in protecting against COVID-19 in solid organ transplant recipients. Importantly, these registry studies lack data on immunogenicity after vaccination in KTRs and can therefore not relate anti-RBD IgG antibody levels to protection against COVID-19 in these patients. Currently, only a few studies investigated whether antibody levels after vaccination are associated with COVID-19 outcomes in KTRs but show inconsistent results [34,35]. The study by Barnes et al. demonstrated that antibody level after vaccination was associated with both incidence and severity of COVID-19 in patients with immune-suppressive diseases, while Hovd et al. demonstrated that it was only associated with severity and not with incidence of COVID-19 in KTRs. The discrepancies between these two studies may be explained by differences in the characteristics of the study population; the nationwide measures that were taken to prevent the spread of COVID-19; adherence to these measures by patients; nationwide infection rates; and most importantly, the timing of infection in relation to vaccination, the number of vaccinations, and circulating variants. This makes it difficult to compare the results of these kinds of studies, especially since the authors did not adjust for these factors in their analyses. Furthermore, these studies did not study vaccine effectiveness by comparing COVID-19 incidence and severity in unvaccinated versus vaccinated patients. How the COVID-19 risk in unvaccinated patients relates to that in vaccinated patients with a low seroresponse after vaccination remains unanswered.

In our large-scale study, approximately a quarter of all KTRs living in the Netherlands sent a blood sample that was collected at home via a finger-prick method to measure anti-RBD IgG antibody levels. In KTRs without previous COVID-19, we confirmed that the immunogenicity of COVID-19 vaccination in KTR is limited. In a previous publication, in which we studied a subset of the present study population in detail, we found that especially mycophenolate mofetil was associated with an impaired immunogenicity [4]. Remarkably, in our study, we found that besides a lower anti-RBD IgG antibody level, the use of azathioprine rather than mycophenolate mofetil was an important contributor to the risk of contracting COVID-19. This may be explained as mycophenolate mofetil being specifically targeted by the adaptive immune system is targeted by mycophenolate mofetil and the innate immune system is targeted by azathioprine So apparently, besides an impaired humoral response after vaccination, whether or not influenced by the use of mycophenolate mofetil, an impaired innate immunity is an important risk factor for contracting COVID-19. Furthermore, we demonstrated that the infection incidence in the overall population is an important factor determining COVID-19 incidence in KTRs.

As, in the Netherlands, nearly all KTRs received COVID-19 vaccination, it was unfortunately not possible to investigate, with sufficient power, vaccine effectiveness by comparing COVID-19 incidence and severity in vaccinated versus unvaccinated KTRs in the same time period. Therefore, we used data from ERACODA, an international database that collects granular information on COVID-19 disease severity, treatment, and outcomes [23]. From this database, we obtained data for the Netherlands from July 2020 to April 2021, the time period immediately before vaccination became available. During this period, the SARS-CoV-2 alpha variant was dominant, whereas during the time period of interest in the present study (April 2021 to December 2021), the delta variant was dominant (Appendix A). It is assumed that the delta variant had a higher transmissibility and induced COVID-19 with similar [36] or even higher severity when compared to the Alpha variant [37,38]. Our data suggest that in KTRs, vaccination results in a reduced risk of COVID-19 hospitalization and mortality when compared to unvaccinated KTRs in the pre-vaccination era, even in patients that remained seronegative after vaccination. This suggests that immunogenicity, other than that expressed as seroresponse, contributes to protection against the disease. In this context, we demonstrated that there is in fact a continuous relationship between the risk for COVID-19, severe COVID-19, and antibody levels, even if the antibody levels are below the threshold for seropositivity. In line with this is our observation that, when we subdivided subjects according to categories of increasing anti-RBD IgG antibodies, we found a dose-dependent efficiency in preventing COVID-19 incidence, even in the lower range. Additionally, vaccination-induced cellular immunity may contribute to protection against disease, which appears to be less susceptible to viral escape and therefore represents a more robust immune response. In contrast to healthy individuals, it has been demonstrated that in KTRs, a cellular immune response can be present in the absence of a positive seroresponse [35,39,40,41]. Nonetheless, the cellular response was closely correlated to the humoral response. Therefore, it can be argued that cellular immune response is not an additional factor associated with the risk of COVID-19, on top of seroresponse.

The findings of this study can probably also be translated to healthy individuals, even though they achieve higher antibody levels after vaccination compared to KTRs [4,26,27,42,43]. It has been demonstrated that the decay in antibody level follows a similar trajectory independent of the peak antibody level shortly after vaccination [44]. On top of that, we have previously shown that this trajectory is similar in KTRs compared to controls [17]. Therefore, we assume that the antibody response, shortly after vaccination, will also be associated with incidence and severity of COVID-19 in healthy individuals. Based on our data, however, we cannot exclude that in healthy individuals, there will be a threshold in antibody level above which no additional protection against COVID-19 will be achieved, as their antibody levels far exceed that of KTRs.

This study was conducted when the delta variant of SARS-CoV-2 was most dominant. It remains unclear whether these results are also applicable for other circulating variants. However, it is plausible that higher antibody levels after vaccination are also important for protection against the current circulating Omicron subvariants. Although antibody levels after COVID-19 vaccination do not reflect the entire immune response after vaccination, it is may be seen as a reflection of the ability of the immune system to respond to vaccinations and infections. Indeed, studies have demonstrated that antibody level after COVID-19 vaccination or at the time of SARS-CoV-2 infection are associated with COVID-19 severity in patients infected with the Omicron subvariants [34,35]. It may well be that this reasoning can be extended to vaccinations against other viral infections, but this needs to be further studied. That some studies could not demonstrate an association between antibody levels after COVID-19 vaccination and the incidence of COVID-19 during the Omicron era [34] may have several explanations. One of these explanations is that with the initial vaccines, targeting the original Wuhan variant, a lower antibody neutralizing capacity against Omicron variants was demonstrated in the general population and solid organ transplant recipients after vaccination [45,46]. Antibodies that formed after an original COVID-19 vaccination may therefore not be able to prevent infection. Now that COVID-19 vaccines are adapted to target current circulating variants, their neutralizing capacity against those variants has improved [47], and probably therewith, the ability to prevent infection. However, this remains speculation and needs to be confirmed by future studies.

Our study has inherent limitations. First, this is an observational study and not a randomized controlled trial. The results should therefore be interpreted with caution. Second, we did not include data on asymptomatic infections, as can be assessed at the end of follow-up by the presence of newly developed nucleocapsid antibodies. In our opinion, this should not be considered a major problem because the aim of vaccination is to prevent the incidence of clinically relevant infections and to improve prognosis of such infections. Lastly, we do not have data available on cellular immunity, which is also important in protection against COVID-19 incidence and severity. Measuring cellular immunity with the gold standard interferon γ (IFN-γ) enzyme-linked immune absorbent spot (ELISpot) assay is, however, labour-intensive and costly and can therefore not be performed in a large-scale study such as ours. Yet, there are other, less laborious and costly assays to measure cellular immunity, for instance the IFN-γ release assay (IGRA) [41]. Although the IGRA is an accurate measure of specific T cell responses in healthy individuals, this may not account for transplant recipients. We previously demonstrated a poor correlation between these two assays in this patient population [48]. Therefore, T cell responses measured in the whole blood of kidney transplant recipients should be interpreted with caution.

The strengths of our study are that this is one of the first real-life studies investigating COVID-19 vaccine effectiveness in KTRs with information about antibody level in a large number of subjects. In addition, we have detailed data about patient and disease characteristics, even about adherence to COVID-19-related lifestyle restrictions and socio-economic status, which are not available in previous studies, for which we could adjust in our analyses. The latter two are important confounders as government-mandated restrictions have been shown to mitigate the spread of COVID-19 [21] and affects the chance of contracting COVID-19. Furthermore, it has been demonstrated that COVID-19 incidence is higher in people with a lower socio-economic status, most likely due to work or living conditions that hamper proper protection. Additionally, severe COVID-19 occurs more often due to a delay in COVID-19 diagnosis and adequate early treatment, which results in a worse prognosis [22].

Although worldwide, the number of hospital admissions and mortality due to COVID-19 is currently low for the general population and for KTRs, the results of our study may have important implications for future vaccination strategies in the case of new pandemic urgencies. Concerns remain that another, more pathogenic SARS-CoV-2 variant will become dominant or that another new pathogenic virus will circulate in the near future for which large-scale (booster) vaccinations will be necessary. An important lesson we can learn from this study is that, in that case, the aim of vaccination should be to reach an antibody level as high as possible to prevent infection and severe disease in this vulnerable patient group and not just reaching seropositivity, as is emphasized in most current studies [31,32,49,50]. For future vaccinations, it should therefore be considered that KTRs do not follow the routine national vaccination programs, but a personalized vaccination scheme instead, based on antibody levels measured at a set time after vaccination. In case a low level is found, repeat vaccination should be considered in that individual because it has been shown that after each new vaccination, on average, higher antibody levels are reached [6,10,51]. In subjects who remain seronegative or at persistently low antibody levels despite repeated vaccinations, alternative vaccination strategies should be developed, for example applying a personalized dose of immune suppressive agents or altering immune suppressive regimens [51,52]. At least these patients should be informed that they are at increased risk of contracting COVID-19 and will have a more severe course in the case of infection. They should therefore be advised to adhere more strictly to preventive strategies, such as social distancing and wearing masks, especially at times with a high incidence of COVID-19 in the general population. In case they contract COVID-19 despite these measures, they should report this to their treating physician, who should then consider giving early treatment dependent on the risk profile and preferences of the individual patient involved in a process of shared decision making [53,54].

## 5. Conclusions

In conclusion, this study demonstrates that in KTRs, antibody level at one month after two vaccination doses is associated in a log-linear relationship with the occurrence and severity of COVID-19 within 6 months follow-up. Our data imply that higher antibody levels should be the aim of COVID-19 vaccination in KTRs, not only reaching seropositivity. Immunosuppressed patients who have no or low antibody levels, measured at a set time after vaccination, should be warned that they remain at risk for COVID-19. Such patients should be offered repeat vaccinations, with or without the use of alternative vaccination strategies.

## Figures and Tables

**Figure 1 viruses-16-00114-f001:**
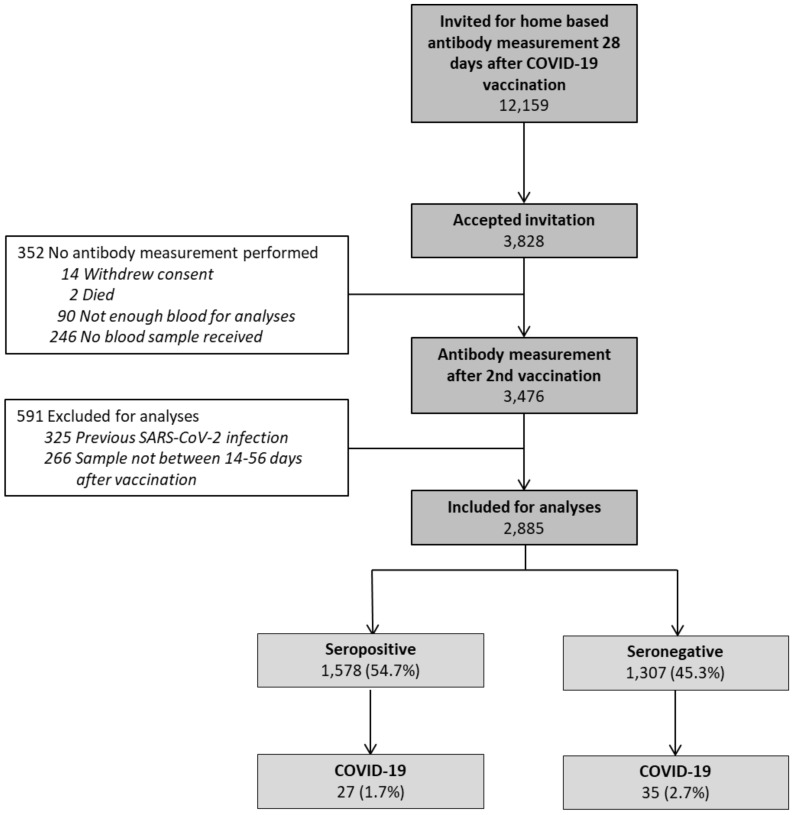
Study enrolment and subject selection.

**Figure 3 viruses-16-00114-f003:**
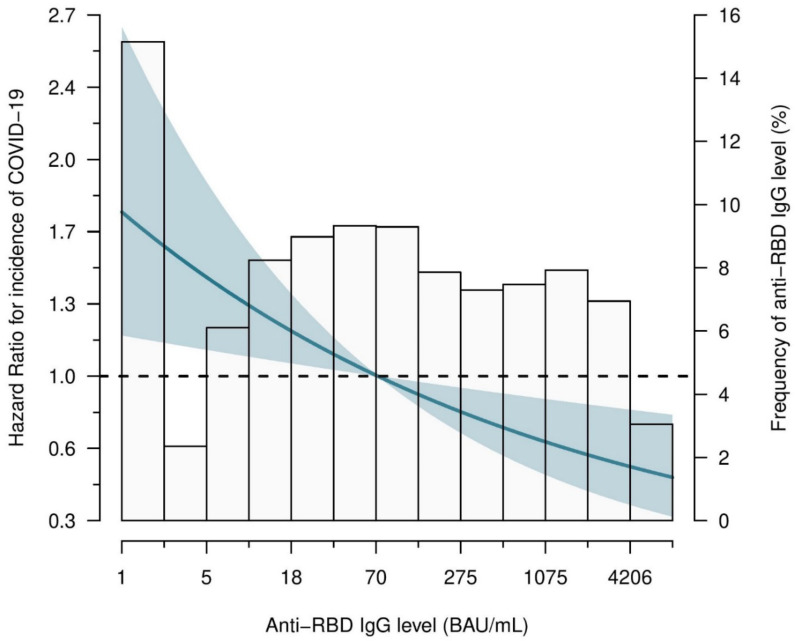
Association between incidence of COVID-19 and anti-RBD IgG antibody level after vaccination. Hazard ratios (HRs) (95% CI) were calculated using Cox regression analysis. The HR is presented as a solid line (restricted cubic spline curve), and the shaded area presents the corresponding 95% CI. The distribution in percentage of anti-RBD IgG antibody level across the study population is shown as bars.

**Table 1 viruses-16-00114-t001:** Baseline characteristics of included kidney transplant recipients overall and according to serostatus after vaccination.

	Overall (*n* = 2885)	Seropositive (*n* = 1578)	Seronegative (*n* = 1307)	*p*-Value
Female, *n* (%)	1214 (42.1)	638 (40.4)	576 (44.1)	0.04
Caucasian, *n* (%)	2514 (92.2)	1373 (91.7)	1141 (92.8)	0.28
Age (years)	57.7 ± 12.9	56.2 ± 13.2	59.5 ± 12.0	<0.001
BMI (kg/m^2^)	26.1 ± 6.3	26.2 ± 5.2	26.0 ± 7.5	0.19
Current smoking, *n* (%)	28 (1.0)	22 (1.5)	6 (0.5)	<0.001
Current alcohol consumption, *n* (%)	1196 (43.1)	692 (45.6)	504 (40.2)	0.004
Number of comorbidities, *n* (%)				<0.001
- 1	1329 (51.4)	818 (53.9)	511 (47.8)	
- 2	631 (24.4)	332 (21.9)	299 (28.0)	
- ≥3	337 (13.0)	169 (11.1)	168 (15.7)	
Comorbidities, *n* (%)				
- Hypertension	2323 (85.3)	1266 (83.4)	1057 (87.7)	0.002
- Diabetes mellitus	549 (21.2)	273 (18.0)	276 (25.8)	<0.001
- History of coronary artery disease	307 (11.9)	155 (10.2)	152 (14.2)	0.002
- Heart failure	158 (6.1)	80 (5.3)	78 (7.3)	0.03
- Chronic lung disease	169 (6.5)	77 (5.1)	92 (8.6)	<0.001
- History of malignancy	113 (4.4)	68 (4.5)	45 (4.2)	0.74
eGFR (mL/min/1.73 m^2^)	51.2 ± 18.7	52.6 ± 19.0	49.1 ± 18.2	<0.001
Primary renal diagnosis, *n* (%)				<0.001
- Glomerulonephritis	455 (20.5)	279 (21.5)	176 (19.1)	
- Interstitial nephritis including pyelonephritis, drug-induced nephropathy, and urolithiasis	155 (7.0)	107 (8.3)	48 (5.2)	
- Cystic kidney diseases	400 (18.1)	233 (18.0)	167 (18.2)	
- Other congenital and hereditary kidney diseases	82 (3.7)	53 (4.1)	29 (3.2)	
- Renal vascular disease, excluding vasculitis	169 (7.6)	100 (7.7)	69 (7.5)	
- Diabetic kidney disease	129 (4.4)	60 (4.6)	69 (7.5)	
- Other multisystem diseases	109 (5.1)	66 (5.1)	49 (5.3)	
- Other	528 (24.9)	292 (22.5)	263 (28.6)	
- Unknown	155 (7.0)	105 (8.1)	50 (5.4)	
Transplant characteristics				
- First kidney transplant, *n* (%)	1770 (85.3)	1077 (85.3)	766 (85.4)	0.94
- Time after last transplantation (years)	7 (3–13)	8 (4–14)	6 (3–12)	<0.001
◦ ≤1 year, *n* (%)	237 (11.0)	118 (9.3)	119 (13.3)	0.004
- Last transplant				
◦ Living, *n* (%)	1368 (63.3)	822 (65.1)	546 (60.9)	0.04
◦ Pre-emptive, *n* (%)	824 (35.8)	492 (36.4)	332 (34.8)	0.44
Number of immunosuppressants, *n* (%)				<0.001
- 1	82 (3.8)	65 (5.2)	17 (1.9)	
- 2	1071 (50.0)	667 (53.4)	404 (45.3)	
- ≥3	988 (46.1)	518 (41.4)	470 (52.7)	
Immunosuppressive treatment, *n* (%)				
- Steroids	1645 (76.8)	975 (78.0)	670 (75.2)	0.13
- Azathioprine	225 (10.5)	187 (15.0)	38 (4.3)	<0.001
- Mycophenolate mofetil	1369 (63.9)	644 (51.5)	725 (81.4)	<0.001
- Calcineurin inhibitor	1767 (82.5)	1026 (82.1)	741 (83.2)	0.52
- mTor inhibitor	167 (7.9)	117 (9.4)	46 (5.2)	<0.001
- Belatacept	19 (0.8)	4 (0.3)	15 (1.4)	0.003
- Anti-IL2-R antibodies	13 (0.6)	7 (0.6)	6 (0.6)	0.97
- Cyclophosphamide	1 (0.04)	1 (0.1)	0	0.36
- Other, not further specified	10 (0.4)	3 (0.2)	7 (0.7)	0.12
COVID-19 vaccination				<0.001
- mRNA-1273	2604 (93.7)	1461 (96.0)	1143 (91.0)	
- BNT162b2	117 (4.2)	44 (2.9)	73 (5.8)	
- ChAdOx1-S	56 (2.0)	17 (1.1)	39 (3.1)	
Adherence to COVID-19 restrictions *	4.25 (3.67–4.67)	4.17 (3.56–4.63)	4.33 (3.75–4.67)	<0.001
Socio-economic status **				0.04
- <−0.2	444 (15.5)	256 (16.3)	188 (14.5)	
- −0.2 to −0.1	257 (9.0)	136 (8.7)	121 (9.3)	
- −0.1 to 0	382 (13.3)	208 (13.3)	174 (13.4)	
- 0 to 0.1	517 (18.0)	275 (17.5)	242 (18.6)	
- 0.1 to 0.2	547 (19.1)	325 (20.7)	222 (17.1)	
- ≥0.2	721 (25.1)	368 (23.5)	353 (27.2)	
Interval between vaccination and blood sample (days)	31 (28–36)	31 (28–36)	31 (28–36)	0.90
RBD IgG antibody level after vaccination (BAU/mL)	72.5 (10.7–600)	495 (142–1716)	8.57 (1.18–21.5)	<0.001
Variables are presented as mean ± SD in the case of normal distribution, as median (IQ interval) in the case of non-normal distribution. or as absolute numbers and percentages in the case of categorical data; due to missing data, numbers and percentages may not match the total number of included patients. *Abbreviations are as follows*: BMI, body mass index; eGFR, estimated glomerular filtration rate. * Adherence to restrictions was determined by the average score for 9 questions on a 1–5 point Likert scale ** Socio-economic status was scored based on financial prosperity, educational level, and recent employment history of households using publicly accessible data from Statistics Netherlands (CBS) [16].				

**Table 2 viruses-16-00114-t002:** Association between risk of COVID-19 and seroresponse in kidney transplant recipients.

	Crude		Model 1		Model 2		Model 3		Model 4 ^1^	
	HR (95% CI)	*p*.val	aHR (95% CI)	*p*.val	aHR (95% CI)	*p*.val	aHR (95% CI)	*p*.val	aHR (95% CI)	*p*.val
Seropositive (yes vs. no)	0.58 (0.35–0.96)	0.03	0.56 (0.33–0.93)	0.02	0.48 (0.26–0.88)	0.02	0.37 (0.19–0.74)	0.005	0.48 (0.27–0.86)	0.01
Age (years)			0.99 (0.97–1.01)	0.23	0.99 (0.97–1.02)	0.61	1.00 (0.97–1.03)	0.81		
Sex (Female vs. Male)			0.94 (0.57–1.56)	0.81	0.93 (0.52–1.67)	0.81	0.98 (0.51–1.85)	0.94		
Diabetes mellitus (yes vs. no)					1.90 (1.01–3.55)	0.045	1.76 (0.89–3.51)	0.11		
Azathioprine use (yes vs. no)					2.87 (1.38–5.97)	0.005	4.04 (1.88–8.69)	<0.001	2.56 (1.25–5.28)	0.01
Adherence to COVID-19 restrictions *							0.93 (0.63–1.40)	0.74		
Socio-economic status **							0.35 (0.09–1.29)	0.12		
Hazard ratio’s (HR) and adjusted HR (aHR) (95% CI) and *p*-values were calculated using a multivariable Cox regression analysis with COVID-19 as the event and days after blood collection as the time variable. * Adherence to restrictions was determined by the average score for 9 questions on a 1–5 point Likert scale ** Socio-economic status was scored based on financial prosperity, educational level, and recent employment history of households using publicly accessible data from Statistics Netherlands (CBS) [16]. ^1^ Model 4 was conducted using a multivariable stepwise backward logistic regression analysis including variables from Model 3, leaving variables with a *p*-value < 0.05.										

**Table 3 viruses-16-00114-t003:** Severity and treatment of COVID-19 in kidney transplant recipients overall and according to seroresponse after vaccination.

	COVID-19			
	All (*n* = 62)	Seropositive * (*n* = 27)	Seronegative (*n* = 35)	*p*-Value
Severity				0.046
- Mild (WHO CPS score 1–3)	47 (75.8)	25 (92.6)	22 (62.9)	
- Moderate (WHO CPS score 4 and 5)	11 (17.7)	1 (3.7)	10 (28.6)	
- Severe (WHO CPS score 6–9)	1 (1.6)	0	1 (2.9)	
- Death (WHO CPS score 10)	3 (4.8)	1 (3.7)	2 (5.7)	
WHO CPS score	2 (2–4)	2 (2–2)	2 (2–5)	0.008
Therapy ^1^				
- Casirivimab + Imdevimab	15 (24.6)	1 (3.8)	14 (40.0)	0.001
- Dexamethasone	14 (19.4)	0	11 (31.4)	0.002
- Tocilizumab	3 (4.2)	0	2 (5.7)	0.22
Time after vaccination (days)	172 (121–201)	167 (115–188)	156 (101–180)	0.29
Variables are presented as median (IQ interval) for continuous data or as absolute numbers and percentages in case of categorical data. *p*-values were calculated using Mann–Whitney U for continuous variables and Pearson’s chi-squared test for categorical variables. *Abbreviations are as follows*: WHO, World Health Organization; CPS, Clinical Progression Scale. * Seroresponse after vaccination was defined as an anti-RBD IgG antibody level ≥ or <50 BAU/mL. ^1^ Due to missing data, numbers and percentages may not match the total number of included patients.				

**Table 4 viruses-16-00114-t004:** Association between COVID-19 severity and seroresponse in kidney transplant recipients.

	Univariable		Multivariable *	
	OR (95% CI)	*p*.val	aOR (95% CI)	*p*.val
Seropositive (yes vs. no)	0.14 (0.03–0.67)	0.01	0.03 (0.001–0.51)	0.02
Age (years)	1.05 (0.99–1.11)	0.12		
Sex (Female vs. Male)	1.29 (0.40–4.15)	0.67		
eGFR (mL/min/1.73 m^2^)	0.93 (0.89–0.99)	0.01	0.93 (0.88–0.99)	0.02
≤1 year after transplantation (yes vs. no)	11.1 (0.89–140)	0.06		
Living donor transplant (yes vs. no)	0.12 (0.02–0.66)	0.02	0.06 (0.01–0.62)	0.02
Socio-economic status **	1.78 (0.13–23.8)	0.66		
Odds ratio’ s (OR) and adjusted OR (aOR) (95% CI) and *p*-values were calculated using a logistic regression analysis. Dependent variable is COVID-19 severity defined as a WHO CPS score of ≥4 or <4, i.e., hospital admission and/or death (yes or no). *Abbreviations are as follows*: eGFR, estimated glomerular filtration rate. * Multivariable model was created using a multivariable stepwise backward logistic regression analysis including all variables from the univariable analysis, leaving variables with a *p*-value < 0.05. ** Socio-economic status was scored based on financial prosperity, educational level, and recent employment history of households using publicly accessible data from Statistics Netherlands (CBS) [16].				

## Data Availability

The data that support the findings of this study are available from the corresponding author, upon reasonable request. Research proposals can be submitted to the consortium members via the corresponding author.

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
