# Peer review of "Incidence and Severity of COVID-19 in Relation to Anti-Receptor-Binding Domain IgG Antibody Level after COVID-19 Vaccination in Kidney Transplant Recipients"

_viruses, 2024, doi:10.3390/v16010114_

Round 1

Reviewer 1 Report

Comments and Suggestions for Authors

 Very interesting paper. SuperB

Reviewer 2 Report

Comments and Suggestions for Authors

The research by A. Lianne Messchendorp and coauthors shows postvaccinal anti-RBD antibody levels to predict COVID-19 development in kidney transplant recipients. The observations were noted in a nationwide study that reinforced the conclusion and could be used for clinical guidelines updates. Moreover, the paper was well prepared, with clear statistics and data-based findings. I want to congratulate you on an excellent manuscript.

I have only minor comments to the authors:

1.    How could your data be correlated with healthy individuals? Please refer to the antibody levels of healthy individuals published previously if possible. Please consider citing: B Lo Sasso, 2022, doi 10.1038/s41598-022-12750-z.

2.      Please expand the thread in the discussion on cellular immunity, pages 5-6, lines 410-418. Please consider citing M Piotrowska Front Immunol. 2022; 13: 832924. doi: 10.3389/fimmu.2022.832924

3.    If possible, it would be valuable to test the impact of CMV on anti-RBD antibody development, as CMV was shown to impact immunity. Please consider Open Forum Infect Dis, 2023 Feb 4;10(2):ofad063. doi: 10.1093/ofid/ofad063. eCollection 2023 Feb.

Reviewer 3 Report

Comments and Suggestions for Authors

The authors studies 2885 kidney transplanted patients for antibody formation after 2nd COVID mRNA vaccination. From their analysis, seropositivity of anti-RBD IgG antibody were associated with a lower risk of occurrence and severity of COVID-19 during 6 months follow-up after vaccination. The antibody levels were also associated with age, sex, diabetes, immunosuppressive drugs, and patients’ life status. The authors concluded that higher anti-RBD-IgG antibody levels should be aimed for in kidney transplanted patients and suggested frequent vaccinations and antibody checks as clinical follow-up.

The authors collected large amounts of samples for analysis. The authors also discussed the limitations of the article as well.

The importance of T-cell activities against the COVID-19 virus has been also pointed out as a factor associated with the severity of COVID-19 infection. If any information related to T cell immunity is present in these samples, it could be good to discuss this (in discussion part).

Minor points

Table 1:

The number of comorbidities is 1 (1-2) for all groups, but p<0.001. Is there something that I do not understand here?  Could authors express what they want to show in some easier way? (Everybody that died also had other conditions? The same in all groups?)

The reason for presenting the "number of comorbidities" and "number of immunosuppressive agents” discretely needs to be explained and if it is the median with IQR in parenthesis this is a confusing way to present it considering the small numbers.

 The table as a whole is difficult to follow and needs clarification and better formatting without center-adjusted text.

Immunosuppressive agents: Others p=0.002, Please indicate which reagents are included. It would be informative to show if ‘others’ include the use of Rituximab or other anti B cell antibody treatment.

Figure 2B

RBD IgG Antibody seronegative criteria is <50 BAU/mL, In the figure authors categorized one group 10-73 BAU/mL. It could be more reasonable to have 10-50, 50-73, or <50 as a reference. Please indicate how many samples are in each group for Figure 2B.

Table 2 is very difficult to read. Please adjust the letter size.
